# Human risk to tick encounters in the southeastern United States estimated with spatial distribution modeling

Rebecca A. Butler[1]*, Mona Papeş[2], James T. Vogt[3], Dave J. Paulsen[1], Christopher Crowe[3], Rebecca T. Trout Fryxell[1]*

**1** Department of Entomology and Plant Pathology, University of Tennessee, Knoxville, Tennessee, United States of America, **2** Department of Ecology and Evolutionary Biology, University of Tennessee, Knoxville, Tennessee, United States of America, **3** United States Department of Agriculture Forest Service, Southern Research Station, Knoxville, Tennessee, United States of America

\* rbutle25@vols.utk.edu (RAB); rfryxell@utk.edu (RTTF)

**Editor:** Álvaro Acosta-Serrano, University of Notre Dame, UNITED STATES

**Data Availability Statement:** Base layers for each map are publicly available from the U.S. census (https://www.census.gov/geographies/mapping-files/time-series/geo/carto-boundary-file.html).

## Abstract

Expanding geographic distribution and increased populations of ticks has resulted in an upsurge of human-tick encounters in the United States (US), leading to an increase in tick-borne disease reporting. Limited knowledge of the broadscale spatial range of tick species is heightened by a rapidly changing environment. Therefore, we partnered with the Forest Inventory and Analysis (FIA) program of the Forest Service, U.S. Department of Agriculture and used passive tick surveillance to better understand spatiotemporal variables associated with foresters encountering three tick species (*Amblyomma americanum* L., *Dermacentor variabilis* Say, and *Ixodes scapularis* L.) in the southeastern US. Eight years (2014–2021) of tick encounter data were used to fit environmental niche and generalized linear models to predict where and when ticks are likely to be encountered. Our results indicate temporal and environmental partitioning of the three species. *Ixodes scapularis* were more likely to be encountered in the autumn and winter seasons and associated with soil organic matter, vegetation indices, evapotranspiration, temperature, and gross primary productivity. By contrast, *A. americanum* and *D. variabilis* were more likely to be encountered in spring and summer seasons and associated with elevation, landcover, temperature, dead belowground biomass, vapor pressure, and precipitation. Regions in the southeast least suitable for encountering ticks included the Blue Ridge, Mississippi Alluvial Plain, and the Southern Florida Coastal Plain, whereas suitable regions included the Interior Plateau, Central Appalachians, Ozark Highlands, Boston Mountains, and the Ouachita Mountains. Spatial and temporal patterns of different tick species can inform outdoorsmen and the public on tick avoidance measures, reduce tick populations by managing suitable tick habitats, and monitoring areas with unsuitable tick habitat for potential missed encounters.

Environmental spatial data used in environmental niche modeling were collected from EarthExplorer (http://earthexplorer.usgs.gov) and EarthData (http://earthdata.nasa.gov/). Tick encounter data can be accessed at Dryad (https://doi.org/10.5061/dryad.v41ns1s3n).

**Funding:** The Forest Inventory and Analysis Program of the Forest Service, United States Department of Agriculture providing funding for the collection and submission of the ticks used in the study; procured by JTV and RTTF. JTV and CC are employed by the USDA Forest Service Southern Research Station. The funders had no role in study design, data collection and analysis, decision to publish, or preparation of the manuscript.

**Competing interests:** The authors have declared that no competing interests exist.

## Author summary

The study highlights the significance of passive tick surveillance data collected by Forest Inventory and Analysis crews in providing valuable information into the presence and distribution of ticks in the southeastern United States. We used ecological niche modeling and generalized linear models to assess the geographic regions and temporal periods associated with where and when ticks are likely to be encountered. In the region, ticks remain active throughout the year with their distribution influenced by climatic and topographical factors. Of interest, maximum temperature was a significant environmental variable for all three species suggesting that distributions may be altered as the climate warms. Elevation and landcover were important variables for both *Amblyomma americanum* and *Dermacentor variabilis*, whereas *Ixodes scapularis* populations were correlated with evapotranspiration, vegetation indices, and soil organic matter. The research also identified new tick occurrence records providing data in a region with minimal infrastructure for tick surveillance, but with many ticks and tick-borne diseases. Continued long-term passive surveillance with collaborations with academic and government partnerships will help monitor tick distribution changes resulting from landscape and temperature changes which affect public health risks.

## Introduction

The geographic distribution of human-tick encounters for medically important tick species is rapidly increasing in the United States (US) [1,2]. Range expansion for commonly encountered tick species (*Amblyomma americanum* L., *Dermacentor variabilis* Say, and *Ixodes scapularis* L.) has resulted in an upsurge of tick-borne disease cases in the last two decades including spotted fever group rickettsiosis, anaplasmosis, ehrlichiosis, alpha-gal syndrome, Powassan virus, and Lyme disease [3–7].

Current distribution range maps for these tick species are commonly based on known occurrences at the continental or county level which is important for surveillance and management of ticks for regional and county health departments [8,9]. These range maps or distribution maps are often based on administrative landmarks (county boundaries) rather than tick and host biological patterns because these maps encompass large spatial areas with environmental factors imprecise for the species' distribution [10]. Environmental or ecological niche models (ENMs) create maps with estimated distributions based on interactions of the species with environmental variables in time and space [11]. Recent ENMs for *A. americanum* predicted this species to inhabit regions in the northern, southeastern, and western regions of the US and Mexico, as well as in the Midwest from eastern Texas to Kansas, Oklahoma, and Missouri [12–14]. Similarly, *D. variabilis* was predicted to occur in parts of Canada and Mexico, as well as northern, southern, and midwestern regions of the US including regions in California [15]. Predictions for *I. scapularis* were distributed throughout regions in the eastern and midwestern US [16]. Understanding the effects that environmental variables have on each tick species distribution is vital because of the recent and predicted impacts of climate and land-use change on their population dynamics [17–20].

Ticks spend the majority of their lives in the environment compared to time spent on hosts; thus, understanding how the environment influences tick populations will lead to increased understanding of ticks and their associated pathogens which can lead to effective management strategies [21]. For example, soil properties such as percent litter coverage and soil moisture are associated with *A. americanum* abundance [22,23]. Additionally, land management

decisions (e.g., burning) were associated with decreased immature *I. scapularis* [24] and *A. americanum* populations [25,26]. Specifically prescribed burns reduce tick abundance by factors such as heat exposure or decrease in soil moisture [26]. Climatic variables (e.g., temperature, vapor pressure, and precipitation) have also been associated with *A. americanum*, *D. variabilis*, and *I. scapularis* (e.g., [27,28]. Landscape variables associated with forests (e.g., land cover, primary productivity) are important for host populations and likely regulate infesting tick populations [e.g., 29,30]. For example, normalized difference vegetation index (NDVI), a measure of greenness, has been associated with the abundance of *A. americanum* [31]. Knowing if an environmental variable is associated with human-tick encounters can be an important surveillance tool for monitoring and a potential management tool for controlling host species.

As tick populations are expanding in a rapidly changing environment, we collaborated with the Southern Research Station's Forest Inventory and Analysis (FIA) Program of the Forest Service, U.S. Department of Agriculture (USDA) to evaluate the environmental conditions that could increase likelihood of encountering ticks. We chose to work with FIA because foresters are known to collect data at a variety of sites with varying environmental conditions and their forest crews are consistently exposed to ticks in the environment. Specifically, foresters collected encountered ticks while working on sites around the southeastern US. We used eight years (2014–2021) of tick encounter data to create environmental niche and generalized linear models to understand where and when ticks are likely to be encountered. Here we test the hypothesis that temporal, climatic, physiographic, and soil variables are reliable predictors of human-tick encounters for *A. americanum*, *D. variabilis*, and *I. scapularis* in the southeastern U.S.

## Materials and methods

### Tick encounter data

Ticks were collected from forest crews employed by FIA. Passive tick collections were opportunistic and occurred when crews worked at plots in the southeastern U.S. between 2014 and 2021. Every year 1/5, 1/7, or 1/10 of the total plots, which are spatially distributed throughout each state (2,000 to 4,800 forest plots per state), are sampled. Crew members visit a single plot each day to inventory each site which takes an entire day. Ticks encountered that day were placed into a single vial containing 80% ethanol and labeled with the date, forestry crew identification number, and GPS coordinates where the crew was working [32]. Encountered ticks were sent to the University of Tennessee, Knoxville Medical and Veterinary Entomology laboratory where they were identified to species and life stage using taxonomic keys [33–36].

### Statistical analysis

Generalized linear models were created from (PROC GLIMMIX) in Statistical Analysis Software (SAS, ver. 9.4, Cary, North Carolina) with two-tailed hypotheses ($\alpha = 0.05$) to analyze how season affects the presence of each tick species together and by life stage. Date of tick collection was transformed into a categorical season variable (winter, spring, summer, and autumn) based on solstice or equinox to account for daylength. Season was used to determine the probability to detect tick presence for each species separately in the binary logit models. Odds ratios and their 95% confidence intervals were calculated for independent variables that were successful at predicting the presence of each tick species. Relative tick encounters, similar to human-tick encounter phenology, were graphed by summing tick abundance for each month per years collected and transformed as a percentage.

## Environmental niche modeling of potential suitability for ticks

FIA crews used hand-held GPS receivers to record the latitude and longitude of each FIA plot. The coordinates used for niche modeling are within 1.6 km (1 mile) of the actual encounter sites because (1) many of the plots are located on private lands and these exact locations are kept confidential (https://www.fia.fs.usda.gov/tools-data/spatial/Policy/default.asp) and (2) ticks may have been encountered traveling to or from plots or while working at the plot. We coarsened the spatial resolution of the model to account for differences between plots and uncertainty of collecting locations. We used these approximate locations as human-tick encounter data to train environmental niche models from 258 sites where *A. americanum* were encountered, 90 sites where *D. variabilis* were encountered, and 36 sites where *I. scapularis* were encountered.

We used 20 environmental rasters (gridded data) as predictor variables in niche models acquired from EarthData (https://earthdata.nasa.gov/) or EarthExplorer (http://earthexplorer.usgs.gov) using NAD 1983 geographic coordinate system (**Table 1**). Each encounter was matched temporally to raster values with the same or nearest date available. We averaged rasters from every date each tick was collected from human hosts to encompass all environmental conditions when tick species were active on human hosts (FIA crews). To compensate for potential spatial sampling errors of ticks, for ticks collected while walking to and from plots and GPS recorded outside property boundaries, we aggregated (coarsened) the original spatial

**Table 1. Spatial variables used in environmental niche modeling for common ticks in the southeastern US.**

| Environmental Variable | Definition | Reference |
|---|---|---|
| Elevation[a] | Height above sea level | [37] |
| Gross primary productivity[b] | Total amount of carbon produced by plants during photosynthesis | [38] |
| Net primary productivity [b] | The difference between carbon dioxide vegetation intakes during photosynthesis and the amount released during respiration | |
| Leaf area index[b] | Quantification of total canopy greenness | [39] |
| Land cover[b] | Surface contents of within the visible landscape | [40] |
| Land surface temperature[b] | Earth surface temperature at a particular location | [41] |
| Evapotranspiration[b] | All forms of evaporation and transpiration | [42] |
| Vegetation indices [b] | A measure of greenness | [39] |
| Precipitation[b] | Condensation of atmospheric water vapor affected by gravitational pull | [43] |
| Vapor pressure[b] | Point at which equilibrium pressure is acquired in a closed container | |
| Minimum temperature[b] | Lowest temperature recorded over a given amount of time | |
| Maximum temperature[b] | Highest temperature recorded over a given amount of time | |
| Burned area[b] | Surfaces which have been affected by burn scar from fire | [44] |
| Living aboveground biomass[b] | Living vegetation above the soil | [45] |
| Living belowground biomass[b] | Living vegetation above below the soil | |
| Leaf Litter[b] | Decomposing plant material on the forest floor surface | |
| Soil organic matter[b] | Portion of the soil consisting of decomposing plant or animal tissue | |
| Dead aboveground biomass[b] | Dead vegetation above the soil | |
| Dead belowground biomass[b] | Dead vegetation below the soil | |
| Hydrologic soil group[b] | Index of the rate that water infiltrates a soil | [46] |

[a] Data were collected from http://earthexplorer.usgs.gov

[b] Data were collected from http://earthdata.nasa.gov/

resolution of rasters from 500 meters to 4.8 km. All raster processing steps were completed in ESRI ArcMap 10.7.

Environmental niche models were fitted using human-tick encounter data and environmental variables in the maximum entropy algorithm Maxent (Version 3.4.0) that estimates species' potential geographic distributions [47]. Maxent was selected due to its robustness in handling presence-only model fitting with limited occurrences [48]. We removed duplicate presence records and ran independent models for each tick species with cross-validation and five replicates, each based on a maximum of 5,000 iterations. For each model, the algorithm selected 10,000 random background samples (or pseudo-absences) to contrast their environmental conditions to those at presence locations. A 10% training presence threshold (allowing 10% of training presence data to be predicted unsuitable) was applied to convert the model outputs of continuous probability of suitability to binary maps of suitable-unsuitable values. Because Maxent is a stable machine-learning platform for controlling correlated variables, environmental raster images were not assessed for autocorrelation beforehand but left to the algorithm to discern [49] (S1 Table). Highly correlated environmental variables do not impact model performance when model transfer is not being used in Maxent [50]. Due to lower sample size for *I. scapularis* only the linear, quadratic, and product features were used in Maxent. Additionally, categorical variables (land cover) were removed from the environmental niche model for *I. scapularis* because the variable contribution was not balanced. For *A. americanum* and *D. variabilis* an addition of the hinge feature was used to build a piecewise linear exponent because there were more encounters (i.e., larger presence dataset) available for model training [47].

To quantify model performance we used area under the curve (AUC) of the receiver operating characteristics; $AUC \geq 0.9$ indicate excellent models, 0.8–0.9 good, 0.7–0.8 fair, 0.6–0.7 poor, and < 0.6 failed models [51]. We also observed the omission error for each model at a 10% training threshold. In addition, we overlaid an U.S. ecoregions boundaries shapefile to evaluate risk based on regional similarities or differences (e.g., geology, soils, climate, vegetation) of distinct geographic areas [52–56].

## Independent model testing

To test each species niche model in addition to the internal cross-validation of Maxent algorithm, we used targeted field work and an independent dataset of previously reported tick records for each tick species by county. We selected three USDA Southern Research Station Experimental Research Forests near the University of Tennessee, Knoxville to confirm the presence or absence of human-tick encounters in this area in 2021. The three experimental forests included Coweeta, Blue Valley, and Bent Creek in western North Carolina and northern Georgia in southern Appalachian Mountains. Each collection site in western North Carolina is representative of forests previously disturbed by logging and mining [57–59]. The Coweeta Hydrologic Laboratory has an area of 1,600 ha and is dominated by mixed-oak forests with understories of noncontinuous mountain laurel and rhododendron [60]. Soils in this region are comprised of Inceptosols and Ultisols [58]. The Blue Valley experimental forest has an area of 526 ha dominated by eastern white pine and stand of oak-hickory in the Blue Ridge Mountains [58]. This area contains acidic, infertile, well drained soils with vegetation primarily consisting of buckberry shrubs [58]. Bent Creek experimental station is comprised of 2,550-ha of upland hardwood forests primarily dominated by oak-hickory stands and understory vegetation of rhododendron [58]. The Asheville Basin region contains soils with low organic matter contents, clay layers, and reduced fertility whereas the Mountain Highland are Inceptisols and acidic [58].

We provided vials to crews to collect encountered ticks, but we also conducted active surveillance in these forests. We set dry ice traps for overnight trapping once monthly (March-June 2021) at five different sites within each forest [61]. The use of dry ice traps were determined by rough terrain at the collection site and limited funding, and the timing of our surveillance was informed by prior knowledge that most ticks are active March through June in the region. Collected ticks were stored in ethanol filled vials, then identified to species and sex as described above.

Additionally, we compared county level records with our predicted distributions. County level records for *I. scapularis*, *A. americanum*, and *D. variabilis* were downloaded on May 14, 2022, from the Center for Disease Control (https://www.cdc.gov/ticks/surveillance/TickSurveillanceData.html). True positives, used to calculate sensitivity, were identified as counties with available tick records and predicted suitable by the niche models. Here, we considered the CDC datasets to be the standard, so false positive counties were those predicted suitable but lacking tick presences, indicating model commission error, and false negative counties were those with known tick presences but predicted unsuitable by the niche models, representing omission error. Although the true distribution for each tick species may not be fully represented due to differences in collection, the CDC datasets represent the most comprehensive comparison for our models because they include surveillance by ArboNET or literature published each year throughout the Southeast. Sensitivity tests were used to determine the proportion of true positives and were calculated in SAS using PROC FREQ. County-level validation of models calibrated with environmental variables with 4.8 km resolution and GPS records of ticks represents a compromise between the need to validate the models and the lack of higher resolution presence data readily available for each species.

## Results

A total of 1,901 ticks were collected and submitted to us during the study period from 384 sites: 1,720 (90.5%) *A. americanum from 258 sites*, 136 (7.2%) *D. variabilis from 90 sites*, and 45 (2.3%) *I. scapularis* from 36 sites. Of the *A. americanum records*, 669 were adults, 851 were nymphs, and 200 were larvae. There were 135 *D. variabilis adults and one D. variabilis* nymph. Finally, there were 45 *I. scapularis* adults. Ticks were primarily collected and submitted by crews in Kentucky, South Carolina, and Tennessee (**Table 2**).

**Table 2. Total number of ticks encountered and submitted by Forest Inventory and Analysis Program crews of the Forest Service, US Department of Agriculture.** Total number of tick-encounter sites are separated by state in the southeastern United States (2014–2021). In total there were 1720 *Amblyomma americanum* (200 larvae, 851 nymph, 310 males, and 359 females), 136 *Dermacentor variabilis* (1 nymph, 73 females, and 62 males) and 45 *Ixodes scapularis* (27 females and 18 males).

| State | *Amblyomma americanum* | *Dermacentor variabilis* | *Ixodes scapularis* |
|---|---|---|---|
| Alabama | 35 (1 site) | 17 (1 site) | 1 (1 site) |
| Arkansas | 15 (13 sites) | 5 (5 sites) | 2 (1 site) |
| Florida | 160 (31 sites) | 3 (3 sites) | 15 (13 sites) |
| Georgia | 4 (3 sites) | 0 (0 sites) | 1 (1 site) |
| Kentucky | 1,030 (121 sites) | 61 (40 sites) | 12 (7 sites) |
| Louisiana | 3 (3 sites) | 3 (3 sites) | 3 (3 sites) |
| Virginia | 7 (4 sites) | 0 (0 sites) | 0 (0 sites) |
| Mississippi | 1 (1 site) | 0 (0 sites) | 0 (0 sites) |
| North Carolina | 9 (3 sites) | 0 (0 sites) | 0 (0 sites) |
| South Carolina | 35 (26 sites) | 13 (11 sites) | 8 (7 sites) |
| Tennessee | 421 (52 sites) | 34 (27 sites) | 3 (3 sites) |
| Total | 1,720 (258 sites) | 136 (90 sites) | 45 (36 sites) |

## Statistical analysis

Season was significantly associated with presence for all three species: *A. americanum* ($F_{3, 349}$ = 13.69; $p < 0.0001$), *D. variabilis* ($F_{3, 349}$ = 6.07; $p = 0.0005$), and *I. scapularis* ($F_{3, 349}$ = 22.14; $p < 0.0001$). The likelihood of *A. americanum* being present was highest in spring compared to any other season, and higher in summer compared to autumn and winter. *Amblyomma americanum* larvae were most likely to be present on human hosts in summer months, nymphs were more likely to be present in spring compared to autumn months, and adults were more likely to be present in the spring compared to any other season. *Dermacentor variabilis* adult encounters were greater in summer months compared to spring and winter months; a single nymph was collected in the spring. Finally, encounters of adult *I. scapularis* were more likely to occur in autumn and winter months compared to spring and summer months. A detailed breakdown of each species monthly percent encounter throughout the eight-year study period is shown in **Fig 1**.

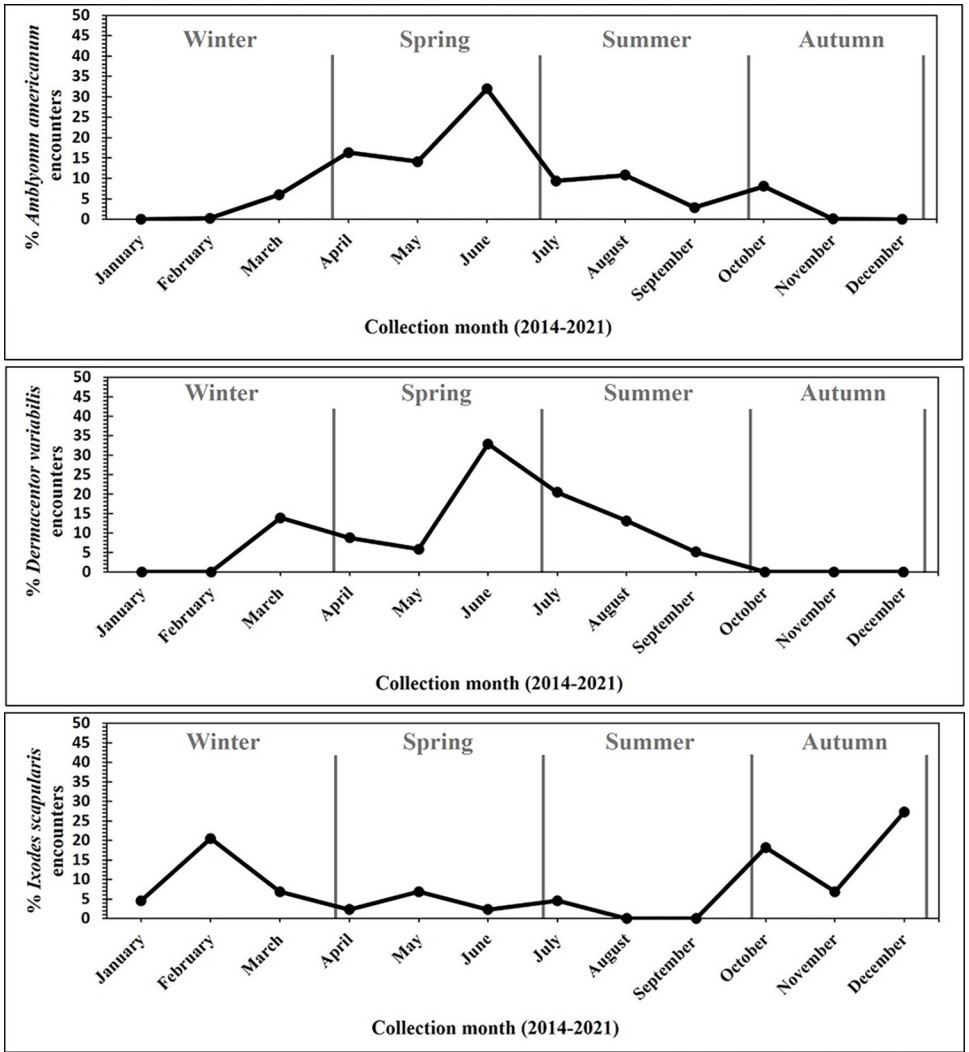

**Fig 1. Relative activity of three species of ticks in southeastern United States.** Percent monthly encounter for *Amblyomma americanum*, *Dermacentor variabilis*, and *Ixodes scapularis* encountered by Forestry Inventory and Analysis crews in the southeastern United States between 2014 and 2021.

**Table 3. Model evaluation and variable contribution to environmental niche models generated with Maxent maximum entropy algorithm for ticks encountered by Forestry Inventory and Analysis (FIA) foresters from 2014–2021.** Values for each variable represent that variables percent contribution to each model, bolded values were those identified as significant contributing variables to the model (greater than 0.1 or 10%).

| Variable | Amblyomma americanum | | | Dermacentor variabilis | | | Ixodes scapularis |
|---|---|---|---|---|---|---|---|
| | model 1 | model 2 | model 3 | model 1 | model 2 | model 3 | model 1 |
| AUC values | 0.83 | 0.85 | 0.80 | 0.87 | 0.85 | 0.86 | 0.82 |
| **Variables Contributing to One or More Model** | | | | | | | |
| Elevation | **16.7** | **20.2** | **15.3** | **24.9** | **21** | **16** | 7.7 |
| Landcover | **19.6** | **16.4** | **16.7** | **21** | **19.2** | **28.8** | - |
| Vapor pressure | **15.4** | **15.4** | **13** | **10.2** | **11.6** | 9 | 0 |
| Precipitation | 6.7 | 5.7 | 6.2 | **11.4** | 9.7 | **10.5** | 0 |
| Maximum temperature | **14.5** | 7 | **14.7** | 16.2 | **17.8** | **10.5** | **11.1** |
| Minimum temperature | 4.5 | **11.2** | 6.9 | 1.3 | 1 | 1.5 | 9.3 |
| Dead belowground biomass | 1.8 | 0.4 | 0.4 | 3.5 | 7.2 | **11.7** | 2.4 |
| Evapotranspiration | 0.9 | 2.4 | 1 | 3.3 | 1.8 | 3.2 | **11.9** |
| Gross primary productivity | 1.2 | 0.1 | 0.4 | 1.5 | 1.7 | 1.9 | **11.9** |
| Soil organic matter | 3.1 | 2.5 | 4.4 | 0.3 | 0 | 0.2 | **24.2** |
| Vegetation indices | 0 | 0 | 0 | 0.8 | 0.5 | 0.1 | **11.6** |
| **Variables Assessed, but Not Contributing to a Model** | | | | | | | |
| Net primary productivity | 5.1 | 2.6 | 6.4 | 2.4 | 1.6 | 1.4 | 2 |
| Hydrologic soil group | 5.8 | 7.4 | 8.5 | 2.6 | 5 | 4.1 | - |
| Land surface temperature | 4.1 | 7.4 | 5.5 | 0 | 0.5 | 0.1 | 2 |
| Litter | 0.2 | 0.6 | 0.2 | 0.2 | 0.3 | 0.1 | 3 |
| Living belowground biomass | 0.1 | 0.6 | 0.4 | 0.4 | 1 | 0.9 | 0 |
| Living aboveground biomass | 0.1 | 0 | 0 | 0 | 0 | 0 | 0.1 |
| Leaf area index | 0 | 0 | 0.1 | 0 | 0 | 0 | 2.8 |
| Burned area | 0 | 0 | 0 | 0 | 0 | 0 | 0 |
| Dead aboveground biomass | 0 | 0 | 0 | 0 | 0 | 0 | 0 |

### Environmental niche modeling of potential suitability for ticks

Lambda file results for all environmental variables in our ENM's can be found in (S2 Table). Three of the five replicate niche models for *A. americanum* had AUC values greater than 0.8 (Table 3). We averaged the raster outputs (probability of suitability) of the three Maxent models to create a single map of potential suitability for *A. americanum* (Fig 2). Model 2 had the highest AUC and elevation, landcover, minimum temperature, and vapor pressure were the best predictors of habitat suitability (Table 3). In models 1 and 3, the same three contributing variables as for model 1 predicted environmental suitability except maximum temperature that had a higher contribution than minimum temperature in these two models. Resulting models indicate potential suitable environments where *A. americanum* are likely to be encountered; thus, crews working in forests throughout Tennessee, Kentucky, northern Florida, Arkansas, North Carolina, and eastern Virginia have an increased likelihood of encountering *A. americanum* (Fig 2).

Three *D. variabilis* models had AUC values greater than 0.8 (Table 2) so we averaged the outputs of the three models to generate the predicted suitability map for *D. variabilis* (Fig 3). High contribution variables to all three models included elevation, landcover, and maximum temperature. Differences between models were represented by precipitation and vapor pressure in the first model, vapor pressure in the second model, and dead belowground biomass and precipitation in the third model (Table 2). Areas of potential suitability and thus concern

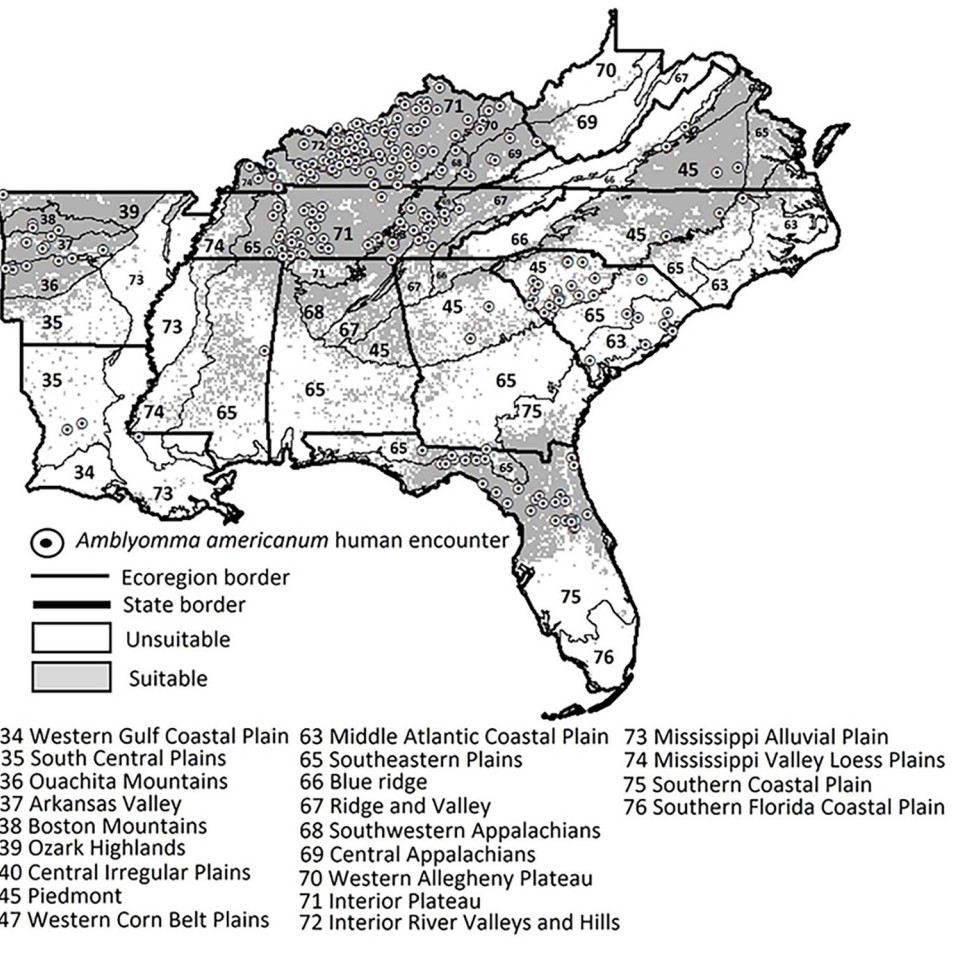

34 Western Gulf Coastal Plain  63 Middle Atlantic Coastal Plain  73 Mississippi Alluvial Plain
35 South Central Plains          65 Southeastern Plains            74 Mississippi Valley Loess Plains
36 Ouachita Mountains           66 Blue ridge                     75 Southern Coastal Plain
37 Arkansas Valley              67 Ridge and Valley               76 Southern Florida Coastal Plain
38 Boston Mountains             68 Southwestern Appalachians
39 Ozark Highlands              69 Central Appalachians
40 Central Irregular Plains     70 Western Allegheny Plateau
45 Piedmont                     71 Interior Plateau
47 Western Corn Belt Plains     72 Interior River Valleys and Hills

**Fig 2. Potential distribution estimated with an environmental niche model for *Amblyomma americanum*.** The model was based on tick encounters collected by Forest Inventory and Analysis Program of the Forest Service, US Department of Agriculture foresters in the southeastern United States (2014–2021). Landcover, elevation, maximum temperature, minimum temperature, and vapor pressure contributed the most to predicting this tick's geographic suitability. The following publically available link was used as the base layer of the map (https://www.census.gov/geographies/mapping-files/time-series/geo/carto-boundary-file.html).

for crews encountering *D. variabilis* included regions throughout Tennessee and Kentucky, northern North Carolina, Alabama, Arkansas, and eastern Virginia (**Fig 3**).

Only one model for *I. scapularis* had an AUC value greater than 0.8 (**Table 2**). The high contribution environmental variables were evapotranspiration, gross primary productivity, maximum temperature, vegetation indices, and soil organic matter (**Table 3**). The area of potential suitability for *I. scapularis* included all ecoregions in the southeastern US; however, there was a reduced risk to encounter *I. scapularis* ticks in southern Florida, eastern Tennessee and Mississippi, and western Arkansas (**Fig 4**).

## Independent model testing

We tested each resulting niche model with data from passive and active surveillance at three Experimental Research Forests and by comparing predicted suitability maps with published or available county-level tick records. Passive and active tick collections at these targeted forests resulted in a total of five *D. variabilis* provided by a camper in the area, but the location of

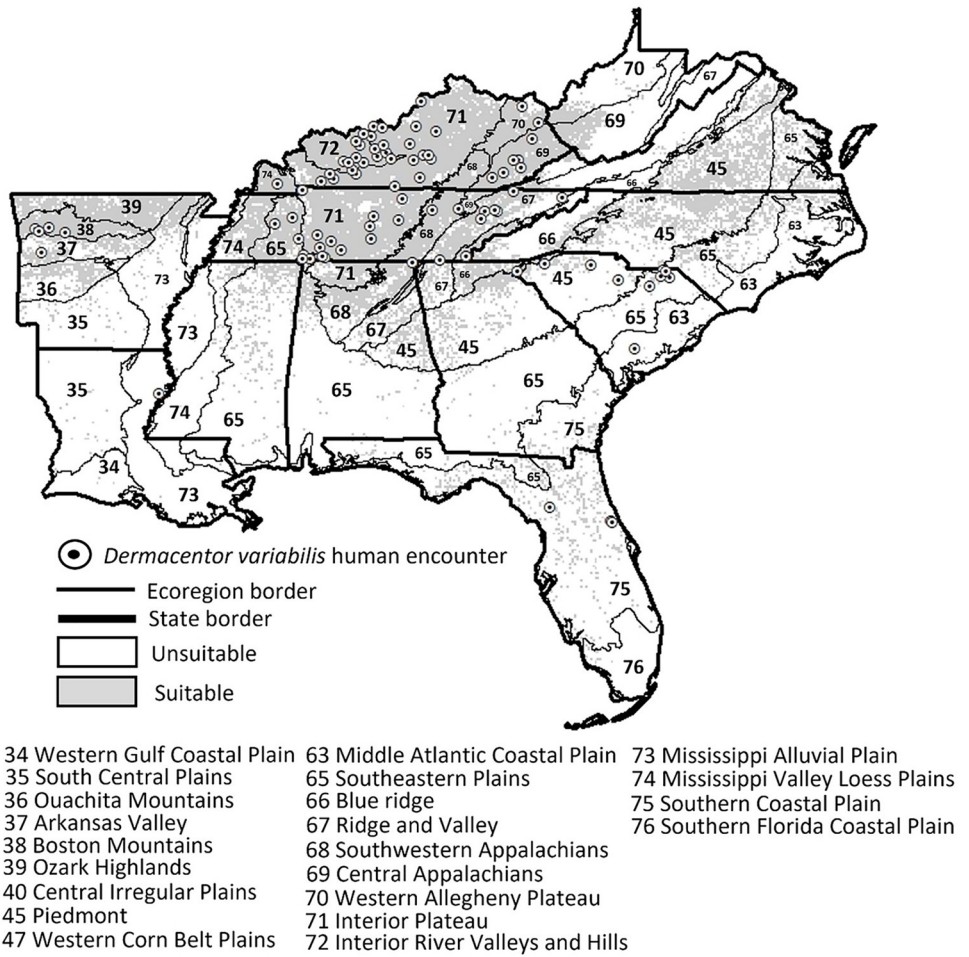

**Fig 3. Potential distribution estimated with an environmental niche model for *Dermacentor variabilis*.** The model was based on tick encounters collected by Forest Inventory and Analysis Program of the Forest Service, US Department of Agriculture foresters in the southeastern United States (2014–2021). Variables landcover, elevation, precipitation, vapor pressure, and dead belowground biomass contributed the most to predicting this tick's geographic suitability. The following publically available link was used as the base layer of the map (https://www.census.gov/geographies/mapping-files/time-series/geo/carto-boundary-file.html).

encounter was not recorded. There were no ticks collected in any of the 15 traps that were placed in the three experimental research forests, nor were ticks encountered on the individual walking to and from each site. The CDC tick surveillance datasets indicated *D. variabilis* to be reported but there were no county records for *A. americanum* and *I. scapularis* in Macon County North Carolina where The Coweeta and Blue Valley Experimental Forests are located. Similarly, CDC surveillance data showed that *D. variabilis* was established but there were no county records for *A. americanum* and *I. scapularis* in Buncombe County North Carolina where The Bent Creek Experimental Forest is located. For all three tick species, our potential distribution maps suggested suitable habitats were present in regions scattered throughout Macon County; however, suitable habitats were found scattered throughout Buncombe County for *D. variabilis* and *I. scapularis* but not for *A. americanum*. Our models correctly predicted sites where dry ice traps were located to be environmentally unsuitable for all three tick

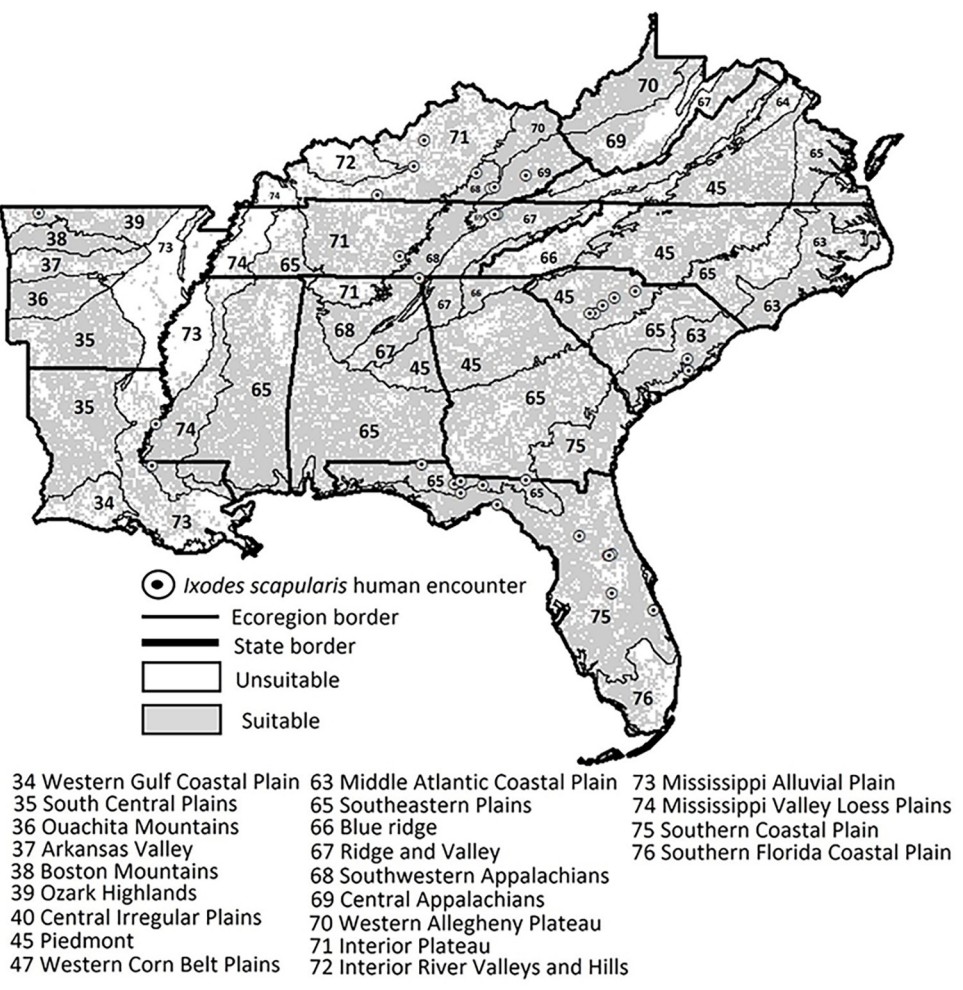

**Fig 4. Potential distribution estimated with an environmental niche model for *Ixodes scapularis*.** The model was based on tick encounters collected by Forest Inventory and Analysis Program of the Forest Service, US Department of Agriculture foresters in the southeastern United States (2014–2021). Variables soil organic matter, vegetation indices, maximum temperature, gross primary productivity, and evapotranspiration contributed the most to predicting this tick's geographic suitability. The following publically available link was used as the base layer of the map (https://www.census.gov/geographies/mapping-files/time-series/geo/carto-boundary-file.html).

species; these absences and unsuitable sites were found in a landscape of predicted suitability by ENMs (**Fig 5**).

Overall, the CDC website identified 390 *I. scapularis*-positive counties and 674 no county records, 718 *D. variabilis*-positive counties and 346 no county records, and 561 *A. americanum*-positive counties and 503 no county records. When we compared our resulting maps at the county level to the previously recorded data, sensitivity values and associated confidence intervals for *A. americanum* were 0.9465 (0.9279–0.9651), for *D. variabilis* were 0.8175 (0.7893–0.8458), and for *I. scapularis* were 0.9956 (0.9905–1.00). This indicates that the models predicted suitable habitats (~encounter areas) similar to those reported by others. There was no observed difference between county status (established, reported, or new county record) and the number of *A. americanum* or *D. variabilis* collected from foresters. Counties noted as established in the CDC database for *I. scapularis* were found to have multiple ticks collected compared to reported or new county records.

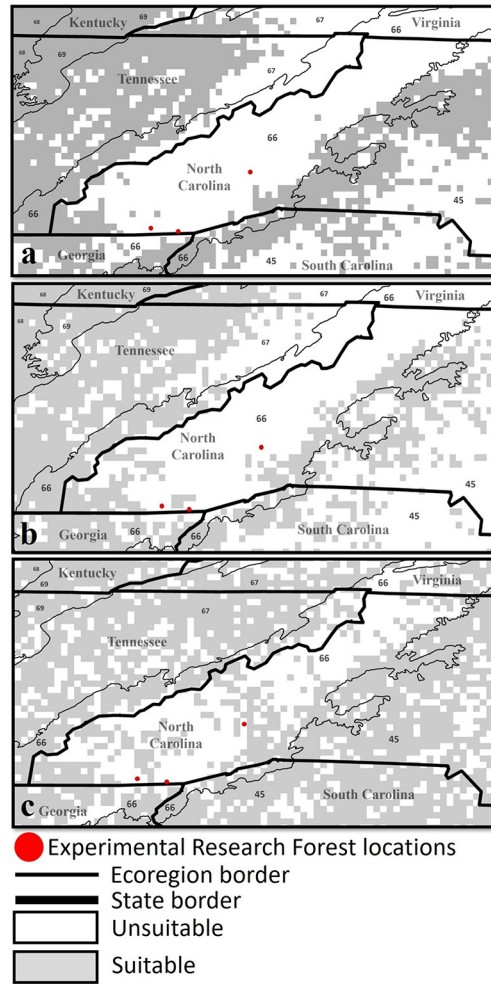

**Fig 5.** Maps indicating unsuitable habitat for sites used as active surveillance with dry ice traps for tick species (a) *Amblyomma americanum*, (b) *Dermacentor variabilis*, and (c) *Ixodes scapularis* in Coweeta, Blue Valley, and Bent Creek experimental forests in western North Carolina and northern Georgia in southern Appalachian Mountains (2021). Eco regions included Piedmont (45), Blue ridge (66), Ridge and Valley (67), Southwestern Appalachians (68), and Central Appalachians (69). The following publically available link was used as the base layer of the map (https://www.census.gov/geographies/mapping-files/time-series/geo/carto-boundary-file.html).

Additionally, as a part of the project, we report several new county records for each species. New county records for *A. americanum* included: Kemper (MS), Madison (FL), Taylor (FL), Gilchrist (FL), Williamsburg (SC), Clarendon (SC), Lexington (SC), Edgefield (SC), Saluda (SC), Newberry (SC), Fairfield (SC), York (SC), Cherokee (SC), Anderson (SC), Halifax (NC), Franklin (NC), Amherst (VA), Henderson (KY), Hancock (KY), Trimble (KY), Kenton (KY), Scott (KY), Carter (KY), Johnson (KY), Perry (KY), Knox (KY), Jackson (KY), Lincoln (KY), Marion (KY), Casey (KY), Adair (KY), Russell (KY), Cumberland (KY), Monroe (KY), Pike (KY) (**Fig 6**). New county records for *D. variabilis* included: Pickens (SC), Wayne (TN), Lawrence (TN), Polk (TN), Daviess (KY), Hancock (KY), Butler (KY), Larue (KY), Green (KY), Taylor (KY), Cumberland (KY), Anderson (KY), Knox (KY), Clay (KY), Anderson (KY), Trimble (KY), Carter (KY), and Johnson (KY) (**Fig 6**). New county records for *I. scapularis* included: West Feliciana (LA), Dade (GA), Union (TN), Simpson (KY), Nelson (KY), Rockcastle (KY), Leslie (KY) (**Fig 6**).

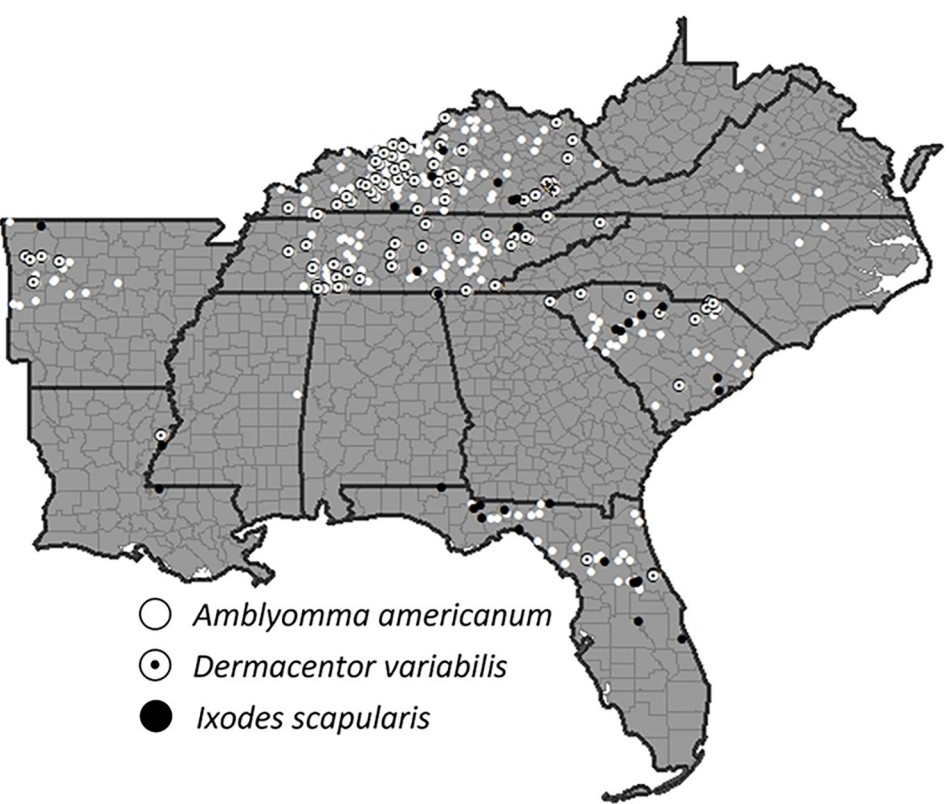

**Fig 6. Sites of tick species collected by foresters in the Forest Inventory and Analysis Program of the Forest Service, US Department of Agriculture foresters in the southeastern United States (2014–2021).** The following publically available link was used as the base layer of the map (https://www.census.gov/geographies/mapping-files/time-series/geo/carto-boundary-file.html).

## Discussion

Three-host ticks reside in the environment for the majority of their two- to three-year life cycle, leaving them susceptible to biotic and abiotic factors [62]. Consequently, ecological niches and habitat suitability often differ for each species spatially and temporally in a rapidly changing environment [63]. In this study, we used ecological niche and generalized linear models to describe geographic regions (where) and temporal periods (when) that represent a risk for humans to tick encounters in the southeastern US. We identified environmental variables associated with the human-tick encounters of three common human biting tick species, which are important indicators for tick encounters and could be targeted for management. The only common environmental variable significantly contributing to the human-tick encounters of all three species was maximum temperature, which supports the expectation that tick distributions will change in warming climates [64–66]. Additionally, long-term passive surveillance is an important tool for obtaining broad-extent occurrence records which can be used to monitor geographic distribution shifts of ticks and public health risk [67,68]. This dataset highlights the potential power of passive tick surveillance by trained scientists from federal agencies to generate accurate distributional information.

### *Amblyomma americanum*

According to many published papers and citizen scientist reports, the most abundant and encountered tick in the southeastern US is *A. americanum* and the species is most likely to be

encountered in the spring [69,70]. The likelihood of encountering *A. americanum* at a site was determined by five environmental variables (elevation, landcover, maximum temperature or minimum temperature, and vapor pressure). Previously, others reported variables such as precipitation, vapor pressure, diurnal range, and temperature as important contributors to predicting *A. americanum* suitability and potential range [17,12,14]. Our environmental niche model highlighted regions at higher risk of encountering *A. americanum* (**Fig 2**) that align with the Interior Plateau, Interior River Valleys and Hills, Ozark Highlands, Boston Mountains, Arkansas Valley, Ouachita Mountains, northern region of the Southeastern Plains, northeastern Piedmont, Northern Southern Coastal Plain, Southwestern Appalachians, and western regions of the Central Appalachian ecoregions. These ecoregions suitable for *A. americanum* contain upland hardwood forests, grassy plateaus, and low elevations in between the Appalachian Mountains and regions in the Southern Plains [71]. A similar pattern to our results was observed based on ticks collected from wildlife hosts in Florida where *A. americanum* was more likely to be present in the northern region of the Southern Coastal Plain [72].

Importantly, our results also predicted when and where *A. americanum* may not be encountered: fall and winter months and locations typically thought to have standing water. Less suitable areas were comprised of wetlands and bottomland hardwoods (e.g., Mississippi Alluvial plain, Southern Florida Coastal Plain), in agreement with previous reports [14,73]. These areas have flat plains with wet soils, marshlands, and swampy land cover [74]. Ticks were less likely to be found in the Southeastern Plains, a region that has historically experienced various land use changes such as plantation forestry and agriculture; these land management decisions could have impacts on tick abundance [75,76]. The Southeastern Plains in Tennessee are known to be occupied by *A. americanum* and their associated tickborne diseases such as human ehrlichiosis and spotted fever group Rickettsioses [77,78].

Similar to our ENM, *A. americanum* populations are less likely to occur in regions with high elevations such as in the Smoky Mountains [17]. Importantly, because *A. americanum* was associated with both elevation and temperature variables, it is possible to investigate these relationships in more detail as higher elevations begin to warm. Future models could predict that, as areas become warmer, more tick encounters will occur at higher elevations where this species was previously absent [17].

### *Dermacentor variabilis*

Foresters frequently encountered *D. variabilis* adults in summer months, as previously reported [79]. Six environmental variables contributed to the environmental niche models with consistent variables including elevation and landcover; and the remaining four variables (maximum temperature, precipitation, vapor pressure, and dead belowground biomass) varied in contribution for each model. Similarly, previous research found precipitation, temperature, and elevation associated with *D. variabilis* [15]. Ticks were most likely to be present throughout Kentucky and Tennessee, and the predicted suitability was fragmented in northern Arkansas, North Carolina, and Virginia. Like our *A. americanum* model, the model for *D. variabilis* predicted suitable areas in ecoregions with upland hardwood forests and fewer extreme elevation changes (e.g., Ozark Highlands, Boston Mountains, Arkansas Valley, Interior Plateau, Interior River Valleys and Hills, Southwestern Appalachians, Piedmont, Mississippi Valley Loess Plain, Ridge and Valley, and the Central Appalachians) [71].

Prior to the range expansion of *A. americanum*, *D. variabilis* was the most commonly encountered tick in southeastern US [14,80–82]. Areas where *D. variabilis* was least likely to be encountered included more southern areas of the study region such as the Southern Coastal Plain and Southeastern Plains, typically characterized as subtropical, low-elevation, sandy, and

areas with marshlands in Florida [71]. Historically *D. variabilis* were reported throughout the Midwest and Southeast but not regions in the upper Northeast and Southwest [83]. Recently, *D. variabilis* encounters were predicted by warm temperatures and low precipitation, thus climate change could improve environmental conditions for this species, increasing their range into the upper Northeast US [15].

The Southeastern Plains region is characterized by drought-prone and nutrient-poor soils, with most land use practices reversing to industrial forests which has resulted in extreme clear cutting [84]. These conditions could lead to unsuitable environmental conditions for this tick species due to its association with precipitation, landcover, and vapor pressure. Of interest is the potential role that dead belowground biomass has on *D. variabilis* because this variable can be manipulated with land and forest management decisions. Intense forest harvesting methods could increase dead belowground biomass due to remaining root structures underground [85]. Dead belowground biomass in the form of dead roots has the ability to hold large amounts of moisture [86] for water storage [87]; which could be important for adult *D. variabilis* survival because they possess greater survival in dry environments compared to other ixodid tick species [88]. Immature *D. variabilis* often quest below leaf litter [89] making them difficult to collect with active surveillance and easier to collect with passive surveillance, from small sized hosts [59,69,77,90].

### *Ixodes scapularis*

Historical encounters of *I. scapularis* in the Southeast are receiving more attention as recent reports are confirming more *I. scapularis* in the region [91]. This study confirms these more recent publications, with adult *I. scapularis* being encountered in the fall and winter and nearly the entire environmentally suitable region [16,92,93]. Although adult *I. scapularis* were encountered less frequently than *D. variabilis* and *A. americanum*, their predicted geographic range was larger. Specifically, *I. scapularis* was predicted to be present in all southeastern states and all ecoregions except the floodplains around the Mississippi River (Mississippi Alluvial Plains and Mississippi Valley Loess Plains). The environmental variable landcover was removed from our *I. scapularis* model because the available presence records did not represent well the variation in landcover categories within the model. Environmental variables that contributed to this ENM included evapotranspiration, gross primary productivity, maximum temperature, vegetation indices, and soil organic matter. Interestingly, the antagonistic effects of transpiration and relative humidity indicate that *I. scapularis* could withstand questing when evapotranspiration is high because it is buffered by drastic changes in relative humidity in forests [94]. Vegetation indices contributing to adult *I. scapularis* populations when they quest in colder months is likely associated with plant dormancy traits [95]. Previous research found variables vapor pressure, elevation, forest cover, isothermality, and temperature associated with their ENMs [16,96]. Similar to our *I. scapularis* model, ticks were less likely to be found at high elevations in the Appalachian Mountains or the Blue Ridge, or southern Florida or the Southern Florida Coastal Plain, and the Louisiana Coast or the Mississippi Alluvial Plain ecoregions [16,71]. In these ecoregions it could be difficult for *I. scapularis* to survive off host because they are dominated by wetland habitats [71].

Studies of immature *I. scapularis* reported southern populations questing in leaf litter while genetically distinct northern populations questing higher on the vegetation, out of leaf litter [91,97]. These behavioral and genetic differences might also be reflected in our ecological niches because variables associated with questing and surviving in an environment (e.g., temperature, evapotranspiration, vegetation indices, etc.) contributed to model calibration. Importantly, the *I. scapularis* submitted by foresters in this study were adults and were not

genotyped. Follow up studies should compare the ecological niches of these genetically and behaviorally distinct populations.

## Model testing

Our models could predict the presence of each tick species but had difficulties predicting the absence of each tick species. This was confirmed with sensitivity and specificity testing as well as conducting targeted surveillance. These results are to be expected because the Maxent algorithm was only trained with presence records that had latitude and longitude coordinates (GPS records). Presence records are readily available for this study because some crews were more vigilant tick collectors and absences could be due to lack of reporting and/or submitting. Nevertheless, our models were able to sufficiently predict tick absence when compared to current literature and active surveillance. Here, few encounters occurred in areas near the foothills of the Appalachian Mountains and our active surveillance supported that as well. Campers in the Bent Creek area helped us confirm *D. variabilis*, indicating ticks may be present in the region, but the likelihood of encounters at these sites is rare compared to other locations, a finding that is similar to other studies in the area [98]. These data support the overall findings that higher elevation areas are often unsuitable tick habitat, which could explain the decreased number of human-tick encounters in high elevation areas like Appalachia [99–101]. Previous research documented that air temperature and density decrease as elevation increases [102]. A possible future research direction could be to monitor these sites as temperature and precipitation change in response to climate change [103], because all three tick species in this study were associated with temperature and none were solely dependent on elevation. Utilizing community science and passive surveillance, this study identified 60 new county records for tick occurrence in eight states. Data generated from this research could be an important tool for raising awareness of increased tick species distributions and used for tick avoidance and management regimes.

## Potential for sampling bias

All three species were absent on human hosts in areas bordering the Mississippi river, coastlines around Louisiana, and southern Florida. This resulted in us reporting that ticks were absent from areas comprising wetlands, analogous to previous studies [99,104]. The absence of ticks on human hosts in these areas could also indicate sampling bias because fewer records were submitted by forest crews working in these areas. This is a known sampling limitation of this study: all data presented here are dependent on forestry crews submitting ticks. Future research should focus on confirming these findings by monitoring ticks in wetland habitats of these regions. Sensitivity and specificity tests indicated our niche models were efficient in their ability to detect known presences of all three species but had poor abilities in detecting a tick absence. Regardless, our relative activity and niche models can be used as a first approximation of risk for encountering ticks and their pathogens, as well as helping with tick management by forestry personnel.

## Land management

Our results indicated that the burned area variable was not a good predictor of human-tick encounters; which suggests that burned locations are not a suitable habitat for tick encounters. Prescribed burning may remove tick and mammal habitats that promote the abundance of tick populations [105], but there are several studies in the Southeast that both support and reject these findings. Previously, prescribed burns reduced the abundance of ticks and the prevalence of tickborne pathogens in an area [106,107] which may be due to increased soil

temperatures, decreased soil moisture, or destruction of tick habitats [108,109]. Conversely, a weak negative association between prescribed burns and the number of ticks in an area could be due to increased host use of burned areas [110] or little impact on tick species that quest and dwell within the soil [111]. Others propose that the methods for controlled burns, such as season, burning intensity, and frequency could be important factors for managing ticks in prescribed burns [112]. Future research to address the potential of controlled burns in large land management areas to reduce recreational and work-related tick encounters is warranted. Tick encounter data can be accessed at Dryad (https://doi.org/10.5061/dryad.v41ns1s3n). [113].

## Conclusions

Passive tick surveillance data provided by FIA crews were valuable in delivering descriptions for when and where ticks were likely to be encountered and contributed to supporting our hypothesis. Data confirmed that ticks are active year-round in the southern US and habitats suitable for encounters can be explained by climatic and topographical features. Regions in the Southeast that were least suitable for all three tick species included the Blue Ridge, Mississippi Alluvial Plain, and the Southern Florida Coastal Plain, whereas suitable regions for all tick species included the Interior Plateau, Central Appalachians, Ozark Highlands, Boston Mountains, and the Ouachita Mountains. Temporal periods associated with encountering each tick species varied; for example, *I. scapularis* was associated with cold seasons (autumn and winter) whereas *D. variabilis* and *A. americanum* were associated with warm seasons (spring and summer). This research provides important information for isolating regions for management of ticks when they are more seasonally active. This study also helps forestry managers alert field crews about tick activity and new county records of tick species. Additionally, this study provides crews with the opportunity to test management decisions against tick encounters. Future studies will aim to combine forest management strategies at high-risk geographic regions to understand associations on human-tick encounters.

## Supporting information

**S1 Table. Correlation values for environmental variables that contributed 10% or more to each environmental niche model.**
(DOCX)

**S2 Table. Lambda file results for environmental variables in each Maxent environmental niche model.**
(DOCX)

## Acknowledgments

We thank the Forest Inventory and Analysis Program of the Forest Service, United States Department of Agriculture for organizing the collection of ticks in this study. This manuscript was prepared in part by United States government employees as part of their official duties and therefore is in the public domain. We also want to thank Jennifer Chandler, Corey Day, and Katy Smith for the initial manuscript review. The Department of Entomology and Plant Pathology at the University of Tennessee helped support RAB's graduate stipend and tuition.

## Author Contributions

**Conceptualization:** Rebecca T. Trout Fryxell.

**Data curation:** Rebecca A. Butler, Dave J. Paulsen, Rebecca T. Trout Fryxell.

**Formal analysis:** Rebecca A. Butler, Mona Papeş.

**Funding acquisition:** James T. Vogt.

**Methodology:** Rebecca A. Butler, Mona Papeş, James T. Vogt, Dave J. Paulsen, Christopher Crowe, Rebecca T. Trout Fryxell.

**Project administration:** James T. Vogt, Rebecca T. Trout Fryxell.

**Resources:** James T. Vogt.

**Supervision:** Mona Papeş, James T. Vogt, Rebecca T. Trout Fryxell.

**Validation:** Rebecca A. Butler, Christopher Crowe, Rebecca T. Trout Fryxell.

**Visualization:** Rebecca A. Butler, Mona Papeş, Rebecca T. Trout Fryxell.

**Writing – original draft:** Rebecca A. Butler.

**Writing – review & editing:** Rebecca A. Butler, Mona Papeş, James T. Vogt, Dave J. Paulsen, Christopher Crowe, Rebecca T. Trout Fryxell.

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
