## [Decision Letter · Decision Letter 0]

11 Aug 2023

Dear Dr Trout Fryxell,

Thank you very much for submitting your manuscript "Human risk to tick encounters in the southeastern United States estimated with spatial distribution modeling" for consideration at PLOS Neglected Tropical Diseases. As with all papers reviewed by the journal, your manuscript was reviewed by members of the editorial board and by several independent reviewers. In light of the reviews (below this email), we would like to invite the resubmission of a significantly-revised version that takes into account the reviewers' comments. 

Reviewers felt that there was considerable value to this study but also were clear that many aspects of this submission need to be improved. The authors are encouraged to carefully consider the comments of the reviewers and ensure that they are adequately addressed in a revision. Also, please be sure to carefully review the submission for spelling and gramma. This is important as PLoS does not copy-edit manuscripts.

We cannot make any decision about publication until we have seen the revised manuscript and your response to the reviewers' comments. Your revised manuscript is also likely to be sent to reviewers for further evaluation.

Sincerely,

Michael R Holbrook, PhD

Academic Editor

Álvaro Acosta-Serrano

Section Editor

Reviewers felt that there was considerable value to this study but also were clear that many aspects of this submission need to be improved. The authors are encouraged to carefully consider the comments of the reviewers and ensure that they are adequately addressed in a revision. Also, please be sure to carefully review the submission for spelling and gramma. This is important as PLoS does not copy-edit manuscripts.

Reviewer's Responses to Questions

**Key Review Criteria Required for Acceptance?**

**Methods**

-Are the objectives of the study clearly articulated with a clear testable hypothesis stated?

-Is the study design appropriate to address the stated objectives?

-Is the population clearly described and appropriate for the hypothesis being tested?

-Is the sample size sufficient to ensure adequate power to address the hypothesis being tested?

-Were correct statistical analysis used to support conclusions?

-Are there concerns about ethical or regulatory requirements being met?

Reviewer #1: -Are the objectives of the study clearly articulated with a clear testable hypothesis stated? Yes

-Is the study design appropriate to address the stated objectives? yes

-Is the population clearly described and appropriate for the hypothesis being tested? Yes

-Is the sample size sufficient to ensure adequate power to address the hypothesis being tested? Yes

-Were correct statistical analysis used to support conclusions? See comments

-Are there concerns about ethical or regulatory requirements being met? No

Reviewer #2: The methods are clearly stated. It is less clear if there are sufficient data for this analysis. The data were collected over 8 years across a wide area. It is unclear if the crews worked equally across the region in all years and all areas. The inter annual variation in ticks could cause the models to provide errant results if the trends are artifacts of an interplay between geography and year. 

There is also little justification for the choice of MaxEnt as the only tool used. There are many other options that could be used for this type of analysis.

Reviewer #3: Lines 105, 305 (and others). The authors are conflating the presence of the tick with the occurrence of a reported human-tick encounter.

Line 124. Why was a binary logit model (tick species present/absent) used for this analysis? Models or tests like a chi-squared test of abundance by season would utilize more of the data and variation temporally. As mentioned below, models/tests should be broken down by life stage.

Line 126. Passive surveillance does not give “true” phenology, but rather the seasonal variation in host-seeking tick-human encounters. This would be modified by things like climatic conditions affecting tick behavior as well as human behavior in tick habitats.

Line 132. Given the small size of larval and nymphal ticks, how confident are the authors that the submitted ticks did not originate from a different FIA plot (>1.6 km from the GPS point of the reported plot)?

Lines 136-138. Given the large variation in numbers of tick species by life stage, the fit ENM may predict highly suitable habitat in places where larvae were collected, just because of their higher numbers relative to other life stages. Were the ENM fit with present/absent or count (# ticks) data?

Line 140. Were the gridded environmental data originally or projected to be on an equal-area projection for fitting the ENMs? Use of non-projected data could violate the equal-area assumption of ENMs since such a large area of land was used in the modeling.

Line 143. Why were averaged rasters across the collection dates used instead of the individual rasters per unique collection? Averaging can obscure the variation in environmental conditions under which a tick was found on a forestry worker.

Line 151. Did you assess auto-correlation between the presence sites?

Line 152. How were pseudo-absence points selected when fitting the models or were background points used? How many points were used?

Line 157. While Maxent is a generally more stable framework in the face of correlated variables than other frameworks (like regression), it is not completely immune to difficulties arising from correlation of covariates (Sillero & Barbosa 2020, https://www.tandfonline.com/doi/full/10.1080/13658816.2020.1798968). How much correlation was present between the covariates included in the model?

Line 191. Why were dry ice traps used for the active surveillance model validations? Dry ice traps are not very effective at collecting Ixodes and Dermacentor ticks and efficacy varies by life stage (e.g., Schulze et al. 1997, doi: 10.1093/jmedent/34.6.615; Falco & Fish 1992, doi: 10.1007/BF01219108). Also, were these traps just run for a single night per month? Given factors like variation in attractiveness of the trap and climatic impacts on tick behavior, lack of collection does not necessarily mean lack of tick presence in these locations.

Lines 172-176. Targeted surveillance only occurred at three sites, and all were negative for the tick species of interest. This seems like too small of a sample size for model validation and only assessed the model’s ability to predict “not suitable” habitat types. Validation of predictions of “suitable” from targeted surveillance seem lacking here.

Line 200. The CDC dataset may not represent the true distribution of presence for each species since tick surveillance is not standardized across the United States. Lack of reported presence in the CDC dataset may represent a lack of surveillance conducted in the county and not necessarily indicate that the tick species is not present in the county. Thus, leading to incorrect estimation of model commission error.

**Results**

-Does the analysis presented match the analysis plan?

-Are the results clearly and completely presented?

-Are the figures (Tables, Images) of sufficient quality for clarity?

Reviewer #1: -Does the analysis presented match the analysis plan? Yes

-Are the results clearly and completely presented? See comments

-Are the figures (Tables, Images) of sufficient quality for clarity? yes

Reviewer #2: The results are clear and make sense. The numbers of ticks collected is just so small spread across this geographic region that is makes the models less useful. The figures are a bit difficult to discern in black and white.

Reviewer #3: The analysis presented does match the analysis plan presented. However, these will need to be updated based on amendments made to the modeling outlines in the Methods.

Table 3. What shape of relationships (linear, quadratic, etc) did these variables have in the models? The methods indicated different forms were trialed across the different models and species.

**Conclusions**

-Are the conclusions supported by the data presented?

-Are the limitations of analysis clearly described?

-Do the authors discuss how these data can be helpful to advance our understanding of the topic under study?

-Is public health relevance addressed?

Reviewer #1: -Are the conclusions supported by the data presented? see comments, but in general yes

-Are the limitations of analysis clearly described? yes

-Do the authors discuss how these data can be helpful to advance our understanding of the topic under study? yes

-Is public health relevance addressed?yes

Reviewer #2: It isn't clear what results add to the understanding of tick populations given the uncertainty of the model data.

Reviewer #3: The conclusions are generally supported by the data presented, but the authors’ language conflates the presence of the tick with the occurrence of human-tick encounters. Data from passive tick submissions provide information on human-tick encounters and not necessarily of the presence of the tick itself given difference in the host-seeking and host preference of each tick species and even life stage. Also, the authors do not make distinctions between temporal variation in tick species presence by life stage, but rather lump all life stages together for each tick species. Abundance and host preference can vary across life stages such that the risk of tick-borne disease can vary across time.

**Editorial and Data Presentation Modifications?**

Reviewer #1: (No Response)

Reviewer #2: (No Response)

Reviewer #3: Lines 57-58. These tick-borne diseases (and syndrome) do not need to be capitalized.

Lines 87, 89. Parentheses and brackets are mixed together for references.

Line 92. The phrase “of host” at the end of the sentence is confusing. Please clarify if you are referring to the tick species or the host species for the ticks.

Maps in figures would benefit from using projected coordinate systems (equal-area projections).

**Summary and General Comments**

Reviewer #1: Major Comments

The use of season as four variables seems very broad as summer encompasses June 21-September 21, in which temperature and life stages of ticks can vary a lot. I wonder if the authors tried to use month rather than season, especially given that the data given in figure 1 is by month not season. There is also the possibility of using week of year. Using AIC could help determine which of those predictor variables are the best, as it is pretty much a given that month would be significant for ticks. But is it biologically significant. We know that spring, summer and fall in general are more likely for ticks to be out, but for human risk, having a more narrower definition of what time of year.

I’m concerned about not using the actual coordinates of the encounter site. Event those the exact locations are kept confidential, as an author works for the USDA FIA and there are no coordinates posted in this MS, 

Can you better explain why you did not assess the raster images for autocorrelation and what was the method for Maxent to discern. Also what does it mean that the land cover variables were not balanced for I. scapularis. Do you have a rationale for why this is acceptable? 

Why was the overnight monthly trapping with dry ice only from March to June?

What exactly are the sensitivity tests used? There are a wide range of test that are labeled as “Sensitivity” and please explain why you are performing these test.

In the discussion section for Ixodes scapularis, can you discuss the inability to include landcover in you ENM model. 

Minor Comments

Line 66-68: The sentence “Environmental niche models…” is unclear. These models do not estimate distribution maps, but create maps with estimated distributions.

Line 71- Remove “potential”

Line 71-73: You use “parts of” twice, which is redundant, You could probably skip the California bit as the sentences primary talks about countries or emphasized the western US instead of just California. 

Line 84: I don’t think you should use a numbered citation in the beginning of the sentence. Put the authors name and place the number where the year would be or rewrite the sentence so that the citation is at the end. Maybe, “Specifically, prescribed burns can reduce tick abundance by factors such as heat exposure or decrease in soil moisture [26]”

Line 86: add “also” between “and precipitation) have” and “been associated with”

Line 99-100: The sentence “In other words… “ is not needed. 

Line 145-146: I’m not entirely sure what you mean by “compensating for potential spatial sampling errors…” and why you would increase from 500 meters to 4.8 km as opposed to 1.6km. 

Lines 269-271: I’m curious if the opposite was true, did you place traps in areas that the model predicted as suitable so as to confirm the model correctly predicted sites where the ticks were found?

Line 275-278: What exactly are these sensitivity values? Are they percentages? What do they tell the reader?

Line 300: You have abiotic twice.

Line 319: You give five variables

Line 350: Foresters frequently encountered…

Line 351-354: Six environmental variables contributed to the environmental niche models with consistent variables including elevation and landcover; and the remaining four variables (maximum temperature, precipitation, vapor pressure, and dead belowground biomass) varied in contribution for each model.

Line 355-357: I’m not completely sure “but” is appropriate in this sentence. Maybe “and”

Line 363-371: Check the tenses in this sentence. There are many instances of using were where was should be used.

Line 378-379: Can you what increasing dead belowground biomass would do and how it will affect ticks?

Line 408: What are these variable that are associated with questing and surviving.

Reviewer #2: Overall, the paper looks at an important concept of understanding tick risk in space. It is unclear if the data are sufficient to support useful findings across the southeastern region.

Reviewer #3: Butler et al. used passive submissions of three tick species from forestry workers in the southeastern United States to estimate the ecological niche of each species and determine spatial and temporal risk of tick encounters. This is an important study to use passive surveillance data to inform risk of human-tick encounters and an interesting population (forestry workers) to use given their unique exposure to tick habitats. I do have some concerns on how the data were used in the analysis and conclusions drawn from these results. Below are my largest concerns.

The authors did not differentiate between tick life stages in the data analysis. The timing of host-seeking varies by life stage for tick species so grouping all life stages together for a tick species obscures the variation in human encounter rates with different life stages. Also, the host preference and density of each tick species and life stage varies, thus skewing monthly percent encounters to more abundant life stages that bite humans. Additionally, the risk of tick-borne disease transmission can vary across life stages of a species (e.g., larval Ixodes scapularis do not transmit Borrelia burgdorferi since this bacterium is not transovarially transmitted), so assessment of risk from the human-tick encounters can vary across the year. Differentiation should be made between life stages for tick species in the analysis, especially the GLM for seasonal differences. 

Throughout, the authors’ language conflates the presence of the tick with the occurrence of human-tick encounters. Data from passive tick submissions provide information on human-tick encounters and not necessarily of the presence of the tick itself given the host-seeking and host preference of each tick species and even life stage. 

More details should be provided for the ecological niche modeling. It appears that correlation between covariates were not taken into account and environmental covariates may have been calculated from unprojected coordinate systems. A list of some common mistakes made in ecological niche modeling (and explanation of how these violate assumptions of the modeling framework) can be found here: Sillero & Barbosa 2020, https://www.tandfonline.com/doi/full/10.1080/13658816.2020.1798968

PLOS authors have the option to publish the peer review history of their article (what does this mean?). If published, this will include your full peer review and any attached files.

Reviewer #1: Yes: Samniqueka Halsey

Reviewer #2: No

Reviewer #3: No
---

## [Decision Letter · Decision Letter 1]

23 Nov 2023

Dear Dr Trout Fryxell,

Thank you very much for submitting your manuscript "Human risk to tick encounters in the southeastern United States estimated with spatial distribution modeling" for consideration at PLOS Neglected Tropical Diseases. As with all papers reviewed by the journal, your manuscript was reviewed by members of the editorial board and by several independent reviewers. The reviewers appreciated the attention to an important topic. Based on the reviews, we are likely to accept this manuscript for publication, providing that you modify the manuscript according to the review recommendations. 

Note from editor: the manuscript is provisionally accepted pending on addressing minor request from reviewer #3.

Sincerely,

Álvaro Acosta-Serrano

Section Editor

Reviewer's Responses to Questions

**Key Review Criteria Required for Acceptance?**

**Methods**

-Are the objectives of the study clearly articulated with a clear testable hypothesis stated?

-Is the study design appropriate to address the stated objectives?

-Is the population clearly described and appropriate for the hypothesis being tested?

-Is the sample size sufficient to ensure adequate power to address the hypothesis being tested?

-Were correct statistical analysis used to support conclusions?

-Are there concerns about ethical or regulatory requirements being met?

Reviewer #1: yes

Reviewer #3: -Are the objectives of the study clearly articulated with a clear testable hypothesis stated? Yes

-Is the study design appropriate to address the stated objectives? Yes

-Is the population clearly described and appropriate for the hypothesis being tested? Yes

-Is the sample size sufficient to ensure adequate power to address the hypothesis being tested? Yes

-Were correct statistical analysis used to support conclusions? Yes

-Are there concerns about ethical or regulatory requirements being met? No

**Results**

-Does the analysis presented match the analysis plan?

-Are the results clearly and completely presented?

-Are the figures (Tables, Images) of sufficient quality for clarity?

Reviewer #1: yes

Reviewer #3: -Does the analysis presented match the analysis plan? Yes

-Are the results clearly and completely presented? Yes

-Are the figures (Tables, Images) of sufficient quality for clarity? Yes

**Conclusions**

-Are the conclusions supported by the data presented?

-Are the limitations of analysis clearly described?

-Do the authors discuss how these data can be helpful to advance our understanding of the topic under study?

-Is public health relevance addressed?

Reviewer #1: yes

Reviewer #3: -Are the conclusions supported by the data presented? Yes

-Are the limitations of analysis clearly described? Yes

-Do the authors discuss how these data can be helpful to advance our understanding of the topic under study? Yes

-Is public health relevance addressed? Yes

**Editorial and Data Presentation Modifications?**

Reviewer #1: Accept

Reviewer #3: I have a few suggestions for the figures. 

Could you add vertical lines (or some such thing) to Fig 1 to provide visual delineation of the (approximate) season definition used in the GLMs?

Could you add to location of the three experimental stations where dry ice traps were placed to the maps illustrating the predicted suitability of the three tick species (i.e., Figs 2-4)? This can aid readers who are not familiar with geography in the southeastern United States and illustrates that these sites were predicted as unsuitable locations by the ENMs.

**Summary and General Comments**

Reviewer #1: The authors have responded acceptably to all of my suggestions/comments from the previous review.

Reviewer #3: The authors adequately responded to my previous comments.

PLOS authors have the option to publish the peer review history of their article (what does this mean?). If published, this will include your full peer review and any attached files.

Reviewer #1: Yes: Samniqueka Halsey

Reviewer #3: No

Figure Files:

Data Requirements:

Reproducibility:

References

---

## [Editor Report · Decision Letter 2]

14 Jan 2024

Dear Dr Trout Fryxell,

We are pleased to inform you that your manuscript 'Human risk to tick encounters in the southeastern United States estimated with spatial distribution modeling' has been provisionally accepted for publication in PLOS Neglected Tropical Diseases.

Best regards,

Álvaro Acosta-Serrano

Section Editor

---

## [Editor Report · Acceptance letter]

9 Feb 2024

Dear Dr Trout Fryxell,

We are delighted to inform you that your manuscript, "Human risk to tick encounters in the southeastern United States estimated with spatial distribution modeling," has been formally accepted for publication in PLOS Neglected Tropical Diseases.

Best regards,

Shaden Kamhawi

co-Editor-in-Chief

Paul Brindley

co-Editor-in-Chief
